# Thermoelectric-Powered Sensors for Internet of Things

**DOI:** 10.3390/mi14010031

**Published:** 2022-12-23

**Authors:** Huadeng Xie, Yingyao Zhang, Peng Gao

**Affiliations:** 1College of Chemistry, Fuzhou University, Fuzhou 350108, China; 2State Key Laboratory of Structural Chemistry, Fujian Institute of Research on the Structure of Matter, Chinese Academy of Sciences, Fuzhou 350002, China; 3Xiamen Key Laboratory of Rare Earth Photoelectric Functional Materials, Xiamen Institute of Rare Earth Materials, Chinese Academy of Sciences, Xiamen 361021, China; 4Fujian Science & Technology Innovation Laboratory for Optoelectronic Information of China, Fuzhou 350002, China

**Keywords:** Internet of Things, thermoelectric generators, fundamental principle, preparation method, application

## Abstract

The Internet of Things (IoT) combines various sensors and the internet to form an expanded network, realizing the interconnection between human beings and machines anytime and anywhere. Nevertheless, the problem of energy supply limits the large-scale implementation of the IoT. Fortunately, thermoelectric generators (TEGs), which can directly convert thermal gradients into electricity, have attracted extensive attention in the IoT field due to their unique benefits, such as small sizes, long maintenance cycles, high stability, and no noise. Therefore, it is vital to integrate the significantly advanced research on TEGs into IoT. In this review, we first outline the basic principle of the thermoelectricity effect and summarize the common preparation methods for thermoelectric functional parts in TEGs. Then, we elaborate on the application of TEG-powered sensors in the human body, including wearable and implantable medical electronic devices. This is followed by a discussion on the application of scene sensors for IoTs, for example, building energy management and airliners. Finally, we provide a further outlook on the current challenges and opportunities.

## 1. Introduction

Thermoelectric materials have been considered potential clean energy sources since their discovery. However, the thermoelectric energy conversion efficiencies of current materials are not high enough to support their large-scale applications in everyday life, so the development of thermoelectric materials in practical applications is limited to some niche fields.

In recent years, many green energy sources have been developed due to the increasing energy demands, among which, photovoltaic technology is one of the most mature choices and preferred solutions. However, in some unattended scenes and indoor circumstances, there is not enough solar energy available, so TEGs are the best solution in specific environments with tiny energy costs.

The Internet of Things (IoT) has emerged with the continuous development of computers, the internet, and mobile communication technology (Figure 1). The IoT, as the ‘Internet where everything is connected’, needs sensors for communication. The radio sensor usually consists of four parts, namely the sensing part, the sending and receiving part, the processing part, and the power supply [1]. Traditionally, radio sensors are usually powered by batteries [2], which in most cases, have limited capacity and may contain toxic chemicals [3]. Moreover, the battery sizes make the installation of the sensor very challenging to maintain in a limited space. Therefore, the premise of the large-scale implementation of IoT in everyday life involves solving the energy supply issues of sensors and meeting the requirements of minimized power supply, which is an essential technology for the next generation of innovative equipment. From a practical point of view, the energy source should be consistent with the operating life of the sensor, which works without an external power supply but a self-powered sensor with renewable energies [4].

Thermoelectric devices, with their miniature designs, can produce orders of magnitude higher energy than the chemical energy produced by the same amount of material; they are the perfect power sources and sensors for IoT applications. Nguyen et al. [5] used thermoelectric material as a power source for a calculator and electronic watch. Zeng et al. [6] considered using thermoelectric sensors in wearable medical devices to detect physiological conditions, such as breathing, pulse, etc. We mainly introduce the application of thermoelectric materials in the IoTs from the aspect of thermoelectric materials as sensors.

This review article will first introduce the history and basic principles of thermoelectric materials (devices). Then, the preparation methods of different thermoelectric materials for thermoelectric generators (TEG) are introduced. After that, we focus on some typical IoT-related applications of TEG, and finally, we shed some light on the future directions and challenges of thermoelectric materials for IoT sensors.

## 2. The Basic Principle of Thermoelectric

The thermoelectric effect that converts heat into electricity was discovered two centuries ago. It includes the Seebeck effect, the Peltier effect, and the Thomson effect [7], constituting the fundamental theories for developing thermoelectric materials. Among them, the Seebeck effect is a phenomenon in that heat energy is directly converted into electricity when a temperature gradient (Δ*T*) is present on both sides of the thermoelectric materials. The Peltier effect is a phenomenon where heat absorption or exotherm occurs when the current passes through the contact of two different materials. Finally, the Thomson effect is when a temperature gradient is applied to a uniform conductor through which a constant current passes, and additional heat absorption or releases will happen.

Materials have different concentrations of free carriers at different temperatures. Therefore, when there is a temperature difference between both ends of the thermoelectric material, the thermodynamically-driven directional diffusion of the electron can form a potential difference, called the Seebeck voltage, as given in Equation: V = S × ∆T. [8]

In general, people use the *ZT* values to evaluate the thermoelectric properties of new materials. That is why the *ZT* value is also known as the thermoelectric figure of merit; the higher the *ZT* value, the better the thermoelectric performance [9]. High-efficiency thermal generators in thermoelectric materials require high *ZT*-type materials. The *ZT* values are expressed as:(1)$$ZT=\frac{S^{2}\sigma T}{k}=\frac{S^{2}\sigma T}{k_{e}+k_{L}}$$
where *S* represents the coefficient, σ represents conductivity, *T* represents absolute temperature, *k* represents thermal conductivity, *k_e_* represents electron thermal conductivity, and *k_L_* represents lattice thermal conductivity, respectively [10]. Thus, the *ZT* value can be increased by increasing the Seebeck coefficient and electrical conductivity or reducing the thermal conductivity [11].

The energy conversion efficiency [12,13] is determined by the *ZT* value, and the temperature difference between hot and cold ends as follows:(2)η=Th−TcTh∗1+ZT¯1+ZT¯+TcTh
where *T_h_* represents the temperature of the hot side of TEG, and *T_c_* represents the temperature of the cold side of TEG. The higher temperature differences can produce higher conversion efficiency and performance output. There are many methods to increase the temperature difference of the thermoelectric leg, among which, optimizing the geometry of the thermoelectric leg can enhance the output performance.

## 3. Preparation Method of Thermoelectric Materials

There are two main categories where the fashions of thermoelectric materials can be applied in a TEG: bulk block materials or thin films. The preparation methods of block thermoelectric materials include vacuum melting, solvothermal, co-precipitation, etc. On the other hand, the preparation of thermoelectric films includes the magnetron sputtering technique, molecular beam epitaxy (MBE), ink printing technology, solution process, chemical vapor deposition (CVD), atomic layer deposition (ALD), and electrodeposition, etc.

### 3.1. Preparation Method of Bulk Thermoelectric Materials

#### 3.1.1. Vacuum Smelting Method

The vacuum melting reaction (Figure 2a) is the most simple and direct synthesis method, which does not require a large number of raw materials. Chen et al. [14] synthesized through vacuum smelting Pb_0.97_Eu_0.03_Te alloy and obtained a *ZT* value of 1.7. However, this approach has some unavoidable shortcomings, such as the need for high temperature and vacuum and antioxidants in the synthesis process; it is hard to obtain chemical uniformity of the product. To reduce the influence of product heterogeneity, after the vacuum smelting, a sintering process is needed to discharge the ions. Zhao et al. [15] used the hybrid technique to prepare Ag_2_Te-doped Cu_2_Te material and achieved a *ZT* value as high as 1.8.

#### 3.1.2. Solvothermal Method

The solvothermal method (Figure 2b) is a widely applied technique due to the advantages of low energy consumption and simple/convenient operations. However, the disadvantage remains that the purity and morphology of the product are difficult to control, and the reaction process cannot be tracked. There are many unsolved details in the interpretation of the principles and the regulation of the experimental process. Wang et al. [9] used the solvothermal method to synthesize n-Bi_2_Te_3_ nanoplates and achieved *ZT* values of up to 1.1, which is a significant improvement so far.

#### 3.1.3. Co-Precipitation Method

The co-precipitation method (Figure 2c) is suitable for industrial mass production due to its advantages of low energy consumption and high efficiency. Ritter et al. [16] successfully prepared Bi_2_Te_3_ powder using this method. However, the co-precipitation method requires complicated steps, such as precipitation, separation, solvent wash, and drying up.

#### 3.1.4. Hot-Pressing Method

The hot-pressing method (Figure 2d) includes pressurized molding, heating, and simultaneous sintering. The advantages of this method are that it can reduce the sintering temperature and shorten the sintering time, and it is easy to obtain the sintering block with near-theoretical density and nearly zero stomatal rates. The disadvantages include short mold life, low production efficiency, and high product costs. In order to improve the mechanical and thermoelectric properties, a YbFe-doped CoSb_3_ skutterudite, which grew by the temperature-gradient zone melting process with a hot press, achieved a superior *ZT* of 0.48 [17].

#### 3.1.5. Spark Plasma Sintering (SPS)

Spark plasma sintering (Figure 2e) is a brand-new technique for preparing functional materials. The advantages of this method are the fast-heating speed, short sintering time, controllable internal structure, energy saving, and environmentally friendly properties. Wan et al. [18] prepared silica limestone (CaSiO_3_) ceramics using spark plasma sintering (SPS) technology with CaSiO_3_ ultra-fine powder as the precursor.

### 3.2. Preparation Method of the Thermoelectric Thin Films

#### 3.2.1. Chemical Vapor Deposition Technology

Chemical vapor deposition technology (Figure 3a) is one of the primary protocols of film preparation due to its advantages of simple operation and accessible raw materials. For example, Rama et al. [19,20] used a new cryochemical vapor deposition method to prepare high-quality, superlattice-based Bi_2_Te_3_ thermoelectric films, which have excellent performance at room temperature, with a *ZT* value of up to 2.4. However, this method is limited by the prolonged deposition rate and the difficulty in achieving large-scale production.

#### 3.2.2. Molecular Beam Epitaxy

The molecular beam epitaxy growth (Figure 3b) is one of the most effective techniques for preparing single-crystal films. By tuning the film growth parameters, it could be a good platform that benefits the performance study. For example, Zhang et al. [21] prepared a series of high-quality single-crystal thin films by this method and studied the relationship between the defects density and properties of the thin films.

#### 3.2.3. Magnetron Sputtering Technique

The magnetron sputtering technique (Figure 3c) has the advantages of fast deposition speed, good film formation uniformity, and easy industrialization. For example, Tao et al. [22] used the magnetron sputtering technique to prepare the Bi_2_Te_3_ thin films with excellent thermoelectric performance. They analyzed the influences of the annealing conditions on the thermoelectric properties of the thin films. Notably, the scrap rate of the target material should be considered to control the cost.

#### 3.2.4. 3D Printing Technology

Three-dimensional printing (Figure 3d) is a rapid manufacturing technology that is simple with low costs. It is possible to construct objects layer-by-layer using molten materials based on digital models. For example, Bian et al. [1] used 3D printing technology to make helical and supporting thermoelectric legs with unique shapes and tested their output performance. They found that the customized legs have higher open-circuit voltage and output power than the conventional legs.

#### 3.2.5. Ink Printing Technology

Ink printing technology (Figure 3e) applies traditional industrial printing technology to the field of thin film preparation, which provides a new road for the industrialization of thermoelectric films. For example, using bottom-up wet chemistry, Saeidi-Javash et al. [23] first synthesized Bi_2_Te_2.7_Se_0.3_ nanosheets, and then they dispersed the nanosheets in a specific solution to make Bi_2_Te_2.7_Se_0.3_ ink. Finally, they used 3D aerosol spray printing to make thin films.

#### 3.2.6. Atomic Layer Deposition Technology

Atomic layer deposition technology (Figure 3f) has attracted wide attention in the field of thin film preparation due to its good film uniformity and accuracy in controlling the composition and thickness of the thin film. For example, by ALD technology, DeCoster et al. [24] prepared PbTe-PbSe thermoelectric films with controlled defect states and excellent performance.

#### 3.2.7. Solution Processing Method

Solution processing is a very conventional method (Figure 3g). For example, Coleman et al. [25] dissolved layered materials, such as WS_2_, BN, and MoS_2_ in appropriate solvents and made films with various casting methods (spin coating, blading coating, etc.).

#### 3.2.8. Electrodeposition Technology

Electrodeposition technology (Figure 3h) is a readily available method for film preparation at room temperature. For example, Kim et al. [26] used this technique to successfully prepare p-type Sb-Te and n-type Bi-Te films for thermoelectric devices.

## 4. Application of Thermoelectric Generator (TEG) in IoT Sensors

Nowadays, about 75% of global energy consumption is based on non-renewable fossil fuels [27]. Many unexpected consequences have emerged after one hundred years of industrialization. The dilemma is that the abrupt reduction of fossil fuels will significantly downgrade the living standard of human society, and the continuous massive use of fossil fuels will further worsen environmental pollution and climate change. As a result, researchers are looking for available clean energy sources to ease the problem. Wind power [28], solar power [29], nuclear fission thermal energy [30], etc., are developed for this purpose. However, due to the inherent shortcomings of solar power and wind energy (requiring the existence of the sun and wind), thermal energy has become one of the relatively stable energy sources considered to be a vital part of solving the global energy crisis [31]. Therefore, the demand for thermal energy conversion technologies is becoming increasingly imperative.

Compared to regular heat engines, such as steam turbines, due to the small sizes and working mechanisms of TEGs, the output can only be in the order of milliwatts to microwatts, which cannot support the energy demand of daily life. However, this energy output is in line with the needs of specific small power equipment, such as the sensors for IoT. Moreover, TEG equipment does not require regular maintenance or replacement of batteries and, therefore, TEGs have long-term power supply capacities that dramatically improve the economic benefits. Wahbah et al. [32] showed that the maximum power output of 20 mW at 22 °C for commercially manufactured TEGs corresponds to a power density of merely 2.2 mW/cm^2^. The study illustrates the worst-case power output of commercial TEGs and the potential of TEG equipment. Since TEGs are light in weight, without emissions and noise, they are more widely used in small electronic equipment [15], such as wearable devices, medical devices, wireless sensor devices, automobile waste heat treatments, and aerospace applications [33] (Figure 1). The following sections will introduce its application in IoT sensors, such as wearables and medical devices, wireless sensor networks, and architectures (Table 1).

### 4.1. Wearable Devices and Medical Equipment

Generally, there is a specific temperature difference between the human body and the ambient temperature, which provides a prerequisite for applying TEGs in wearable devices and medical devices. The human body, as with any object, can obtain and lose heat through conduction, convection, and radiation. Specifically, they include the conduction between the contact objects and/or substances; convection involving the transfer of heat from the warm body to the air above or inside the body, where blood, gas, and other fluids are the media; and radiation where heat transfers/exchanges between the surface of the human body and the surrounding environment. These three effects work together in most cases [38]. Wearable devices and medical equipment that can work using the power provided by the temperature difference could be important parts of IoT.

When using TEG as the power supply for wearable equipment, in addition to the requirement of conversion efficiency, one has to consider additional requirements, such as biocompatibility, wear resistance, toxicity, flexibility, and so on. So far, Bi2Te3 and its derivatives are the most commonly used low-temperature thermoelectric materials, showing high thermoelectric performance around room temperature. Even with the advantage of mass production, they suffered from poor mechanical properties and toxicity of telluride, which limits their application in some wearable or implanted devices. In general, the currently popular thermoelectric materials are not yet satisfactory for powering wearable sensor devices, considering their costs, limited large-scale availability, and high quality. For in-vitro application, Lu et al. [39] deposited the nanostructured telluride on silk fabric to produce a flexible thermoelectric material, which could effectively convert body heat energy into electricity. The reduced direct contact with the human body by changing the position of telluride on fabrics can reduce harm to the human body.

Flexible thermoelectric materials show broad prospects in wearable thermoelectric materials. To avoid the low performance of organic thermoelectric films, Cao et al. [40] prepared flexible thermoelectric films by sputtering Sb (p-type) and Ni (n-type) films on the polyimide substrates. Since transparency is essential for wearable devices. Wang et al. [41] prepared TEG with p-n junctions formed by p-type PEDOT: PSS and n-type indium tin oxide (ITO), respectively, to obtain transparent flexible thermoelectric films with high conductivity and the Seebeck coefficient.

Perovskite materials have attracted much attention in thermoelectric applications due to their low thermal conductivity, high carrier mobility, and the Seebeck coefficient [42]. Ye et al. [43] developed organic–inorganic lead halide perovskite single crystal (MAPbI_3_)-based TE devices and found that improving electrical conductivity is the key factor to realizing the high thermal electric performance of perovskite.

In this regard, highly efficient TEGs that can produce tens or hundreds of microwatts in the presence of such temperature differences are highly desired. In addition, to making the wearer comfortable and suitable for mass production, some basic rules must be followed [44]:(1)TEGs cannot be designed independently of their environment, and they must meet the thermal matching conditions between the heat source (person) and radiator (ambient air);(2)The thermal resistance and heat source of the thermal reactor and the environment should be as exact as possible to achieve the maximum output power;(3)Generate enough voltage to supply power to the electronic device;(4)Wear comfortably;(5)The size should be as small as possible.

#### 4.1.1. Wearable Devices

Wearable electronics often require stable, reliable, and durable energy sources [45]. The human body is a stable and constant heat source, and the heat generated by a person during the day is enough to support most portable electronic devices. Wearable TEGs can constantly convert heat from the human body into electricity [8], and in conjunction with medical devices or GPS, they could be used as inpatient autodetection watches, wildlife locators, and so on; this could be seen in recent past years via wrist-wearable thermoelectric devices [35] (Figure 4a–c), thermoelectric wrist-powered thermoelectric watches [34] (Figure 4d,e), and thermoelectric watches [46] (Figure 4f).

When TEGs are used on wearable devices, there are unresolved problems, such as poor mechanical performances, the inability to self-repair, and the complicated utility of devices. Recently, Ren et al. [47] introduced a high-performance wearable TEG with self-repair, recyclable, and Lego-style reconfiguration capabilities. Both the ideas and novel designs pave the way for the application of thermoelectric material in the IoTs.

#### 4.1.2. Medical Equipment

In this rapidly developing human society and due to the continuous improvement of modern medical systems, the shortcomings of traditional mobile medical equipment have become increasingly prominent, for example, limited cycle numbers for the batteries used in the equipment, the high costs of the regular replacement of batteries, and challenging real-time monitors for patients. The use of TEGs as the power supply can alleviate these problems to a large extent. When used in the medical field, TEGs can be divided into implantable and external types, placed in and outside of the body (in most cases, on the surface of the skin) (Figure 5).

The human body can be roughly divided into three types of regions—muscle, fat, and skin areas. To figure out the area that is the most suitable for TEG applications, Yang et al. [49] demonstrated through a series of systematic studies that the highest temperature gradient occurs when the device is placed near the human skin surface of the body. When the maximum temperature difference is established, it can be the best candidate position for implanting and locating the TEG. Nevertheless, the technology is not yet perfect, and biocompatibility and insulation problems are still not completely solved.

In contrast, the external placement of the equipment is safer and more convenient. Lay-Ekuakille [38] invented two battery-free wireless dual-channel electroencephalography (EEG) systems. (Figure 6a,b) The EEG headband over the electrodes could be put on the scalp and record the brain’s electrical activity. This can be used as a noninvasive clinical tool to assess brain functioning, especially for patients with epilepsy, sleep disorders, etc. From the frequency distribution of brain waves, it can identify organ functions or diseases. On the other hand, Hoang et al. [50] implemented a TEG-based thermal energy collection system that could take energy from the human body to power the sensor nodes that detect the fall-down events of patients or elderly people. Torfs et al. [51] prepared a wireless pulse oximeter for non-invasively measuring oxygen levels in human blood, and the device is powered entirely by a watch-like TEG (Figure 6c). Similarly, Bavel et al. [44] designed a biomedical hearing aid powered by body heat (Figure 6d). If this smart design can be popularized, it is good news for the vast majority of people who are deaf and people with hearing impairments. In a similar manner, a TEG-powered portable EEG system [52] can monitor the heart data in cardiovascular patients and remind healthcare workers to take action if the heart function is abnormal.

The integrated medical system in clothing can be applied widely and it has market prospects due to its incomparable convenience of use. Leonov et al. [53] integrated a TEG-powered pulse oximeter, an EEG system, and an electrocardiogram (ECG) system into a shirt, which not only meets the needs of daily wear but also provides particular medical care for some specific patient groups. Lu et al. [39] creatively combined commercially available silk fabrics with TEGs using traditional printing technology to harvest human body energy. The designed patterns provided adequate functionality without affecting the aesthetics.

### 4.2. Wireless Sensor Networks

Nowadays, wireless sensors can be found in every aspect of life, including industrial and agricultural monitoring systems, wearables, medical devices, etc. Furthermore, a single wireless sensor has been replaced by wireless sensor networks, which are similar to a neuron network in a specific area playing the role of real-time monitoring.

Due to the rapid development of the internet, the rapid transmission and change of information provided an opportunity for developing the next-generation power supply for the wireless connection. For example, Georg et al. [36] combined TEGs with the Bluetooth systems of mobile phones, allowing mobile phone users to receive the weather conditions in real time, which significantly changed the lag of the current mobile phone weather forecast information transmission and improved user experience.

Recently, wireless sensor networks have been considered for various aviation applications in sensing, data processing, and wireless transmission of information. Dilhac [37] thoroughly introduced the installation position of TEG in aviation applications. In addition to the traditional application of the generator and exhaust system, etc., they can be included in the hot positions in the cabin, the outer surface (external air exchange, cruise altitude changes will produce the instantaneous thermal gradient, mainly in the take-off/climb and descent/landing stages), and on the rear hanger fairing. Furthermore, sensors on the fuselage can monitor the mechanical load during the operation to achieve early detection of the fatigue of materials. TEG-powered sensors not only solve the cumbersome wiring problems of traditional sensors but also alleviate the battery instability in extreme environments and the high cost of regular battery replacement.

### 4.3. Architecture Field

There has been a long history of TEGs being used in the field of architecture, and they have been commercialized, as shown in Figure 6a. The buildings are full of populations who live or work inside and, therefore, they can produce large amounts of heat losses, which are opportunities for TEGs. The thermal qualities of general buildings are excellent, so their temperatures change very slowly. Therefore, when there are temperature fluctuations in the weather, these fluctuations may lead to local temperature gradients that allow the TEG-powered sensors to operate. Wang et al. [54] believed that many high-voltage AC units, water heaters, boilers, hot water pipes, and other heating units existing in modern buildings are all potential heat sources. For example, Lezzi et al. [55] designed and implemented a radial TEG integrating thermal pipe insulation, which can power the sensor circuits with wireless transmission capacity and make good use of the waste heat generated by the hot water pipeline (Figure 7b). The waste heat in buildings is tremendous; if they could be effectively exploited, this may somehow alleviate the energy supply problem.

In conclusion, wearable devices powered by mini-TEGs for monitoring the medical or physical conditions of patients need to form wireless sensor networks to realize real-time detection and diagnosis of the status of the object. Furthermore, by using the TEG as the energy supply for the signal transmission and reception unit, the advantage will be the ability to reuse the widely available low-grade thermoenergy from the human body or building, which can alleviate the extra cost from the external energy supply. A summary of TEG applications for IoT is listed in Table 2.

## 5. Conclusions

In summary, this paper first outlines the basic principles of thermoelectric materials and different preparation methods of thermoelectric devices. Due to their lightweight emissions and noise-free properties, TEGs have gradually entered the public view and are used in the field of wearable devices, medical devices, wireless sensor networks, and architecture fields. Of course, beyond the application areas of thermoelectric materials mentioned above, they have also been used in environmental monitoring, civil infrastructure, national defense, manufacturing, and other applications [2]. As a result, research on high-performance thermoelectric materials has become a hot topic, and people use these materials to create multi-functional sensors for various purposes (Figure 8).

## 6. Perspectives

The performances of thermoelectric materials are not yet perfect, and they will face opportunities and challenges when people try to push the thermoelectric performances even higher and find more applications in serving human beings.

Novel materials could be invented to enhance the *ZT* of thermoelectric materials further. Interestingly, various materials with superior *ZT* have been discovered. For example, SnSe [56,57,58,59,60] and Mg-based materials [61,62]. Moreover, in specific scenarios, the importance of flexible thermoelectric materials is gradually being reflected, and the rapid development of organic thermoelectric materials provides a prerequisite for it [63,64,65,66].

Although commercial TEGs can be used to build self-powered sensor nodes, they are pretty expensive and show only limited power maxima in small-sized devices. It is proposed that one should further increase the number of thermocouples and reduce the height of thermocouples within the technical limits [51]. Watanabe et al. [67] designed a new micro-TEG device that is driven by temperature gradients perpendicular to the substrate without any cavity structure below the thermocouple, thus shortening the height of the thermocouple and increasing the thermoelectric power.

Integrating thermoelectric equipment with photovoltaic equipment is also a good idea [68]. The addition of photovoltaic equipment solves the problem of an insufficient power supply of thermal power equipment, can help the device maintain a specific temperature difference, and improve the stability of the photovoltaic equipment. For example, the photothermal power devices prepared by Jin et al. [69] showed significant photothermal conversion capacity and very low thermal conductivity. So this device can maintain a specific temperature difference (20 K) without external auxiliary means, which is also the premise for applying TEG.

New thermoelectric materials with higher intrinsic thermoelectric properties for more application scenarios should be developed. For example, we are currently pursuing studies on multi-functional sensors based on thermoelectric materials, which are, however, restricted by limited preparation methods and complex structural designs [70,71]. With technological advances, we believe that these problems could be solved by more high-performance thermoelectric materials.

## Figures and Tables

**Figure 1 micromachines-14-00031-f001:**
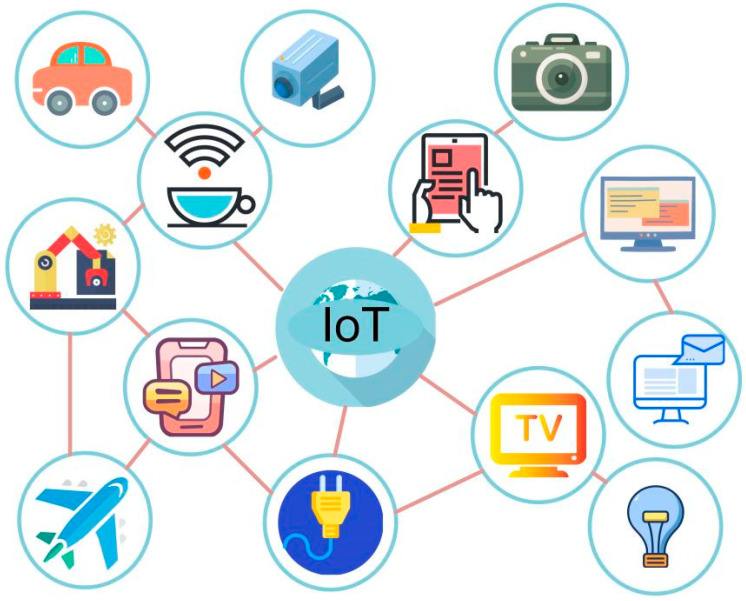
The application of thermoelectric-based sensors in the IoT of daily life.

**Figure 2 micromachines-14-00031-f002:**
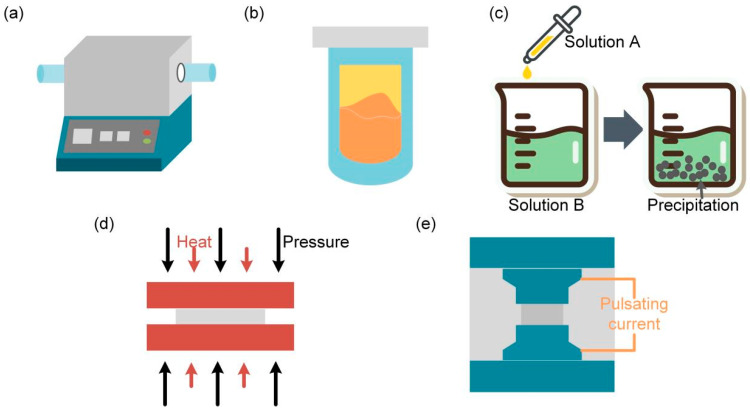
The synthesis method of bulk thermoelectric material. (**a**) Vacuum smelting method, (**b**) solvothermal method, (**c**) co-precipitation method, (**d**) thermal pressing method, (**e**) spark plasma sintering.

**Figure 3 micromachines-14-00031-f003:**
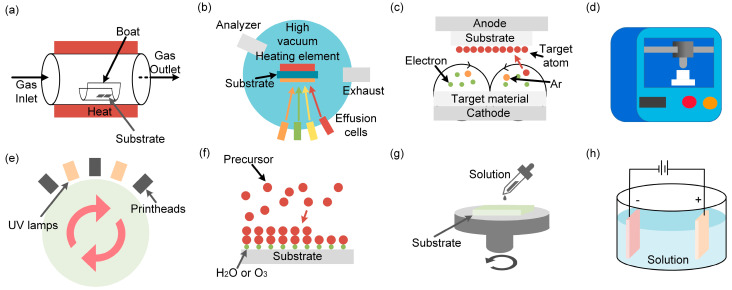
The preparation method of the thermoelectric film. (**a**) Chemical vapor deposition technology, (**b**) molecular beam epitaxy, (**c**) magnetron sputtering technique, (**d**) 3D printing technology, (**e**) ink printing technology, (**f**) atomic layer deposition technology, (**g**) solution processing method, (**h**) electrodeposition technology.

**Figure 4 micromachines-14-00031-f004:**
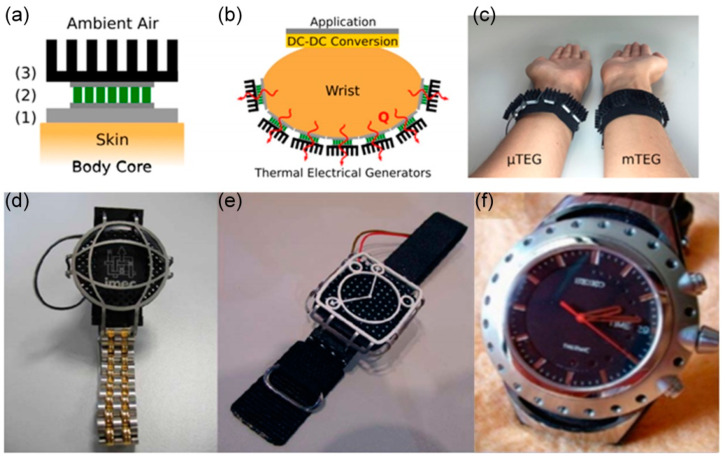
(**a**) Schematic diagram of the heat collector module on the human body: (**1**) the thermal interface of the heat source/skin, (**2**) the TEG used for thermal power conversion, and (**3**) the radiator emits the heat into the ambient air. (**b**) Schematic diagram of the wrist body thermal drive wearable devices, including an energy collector, DC–DC conversion and storage, and application circuits. (**c**) Pictures of the collected wrist bands: mTEG (macroscopic thermo-legs fabricated by classic fabrication technology) and μTEG (a higher number and density thermo-legs produced with micro-fabrication techniques). Reprinted with permission from [35]. Copyright 2017, Elsevier. Design examples of wearable TEG. (**d**) A waterproof version of the “Mini-Matrix 2R” containing 24 thermoelectric reactors; (**e**) a “five to seven” design with 32 thermoelectric reactors and a pin radiator. Reprinted with permission from [34]. Copyright 2007, IEEE-INST ELECTRICAL ELECTRONICS ENGINEERS, INC. The metal electric shock protection grid above the TEG does not contact the radiator. (**f**) Photo picture of thermoelectric watches. Reprinted with permission from [46]. Copyright 1999, IEEE-INST ELECTRICAL ELECTRONICS ENGINEERS, INC.

**Figure 5 micromachines-14-00031-f005:**
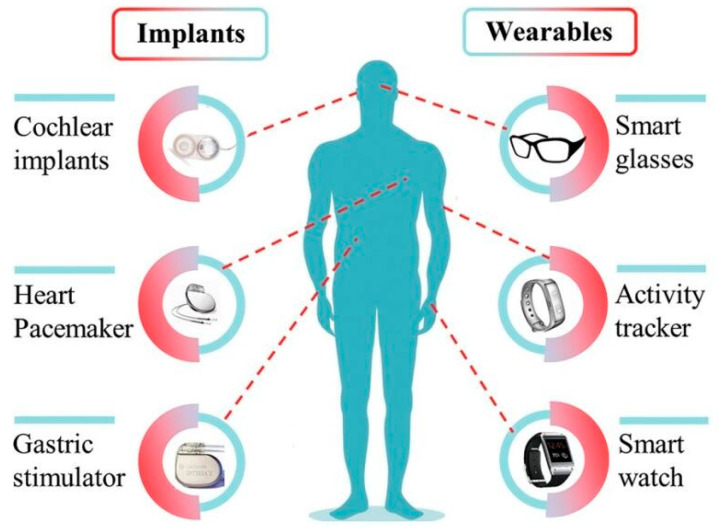
Some application cases of small thermoelectric devices in wearable and medical areas. Reprinted with permission from [48]. Copyright 2019, Wiley-VCH.

**Figure 6 micromachines-14-00031-f006:**
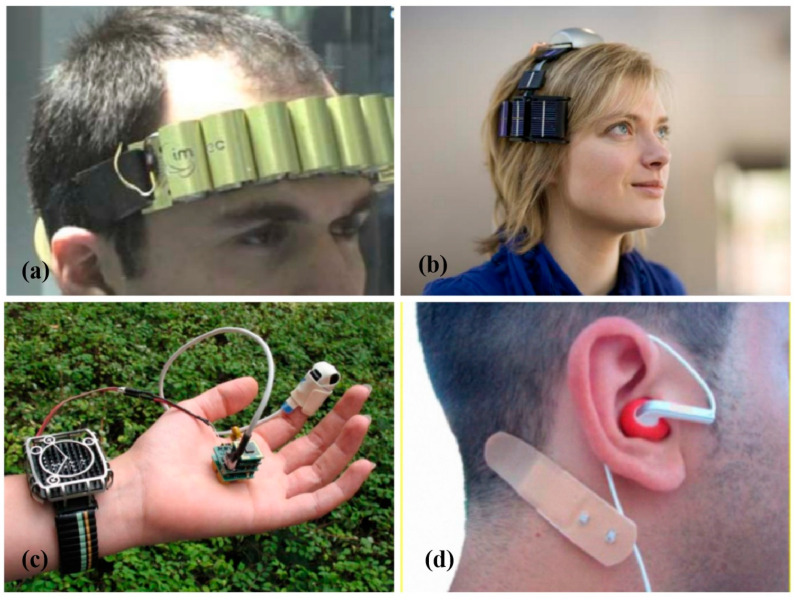
(**a**) Thermal power supply wireless EEG on the headband. (**b**) Wearable headset-type wireless EEG system with hybrid power supply. Reprinted with permission from [38]. Copyright 2009, IEEE-INST ELECTRICAL ELECTRONICS ENGINEERS INC. (**c**) “Watch” type TEG (left), an electronic device with a wireless module (center), and a commercial finger pulse oximeter (right). (**d**) Biomedical hearing aids.

**Figure 7 micromachines-14-00031-f007:**
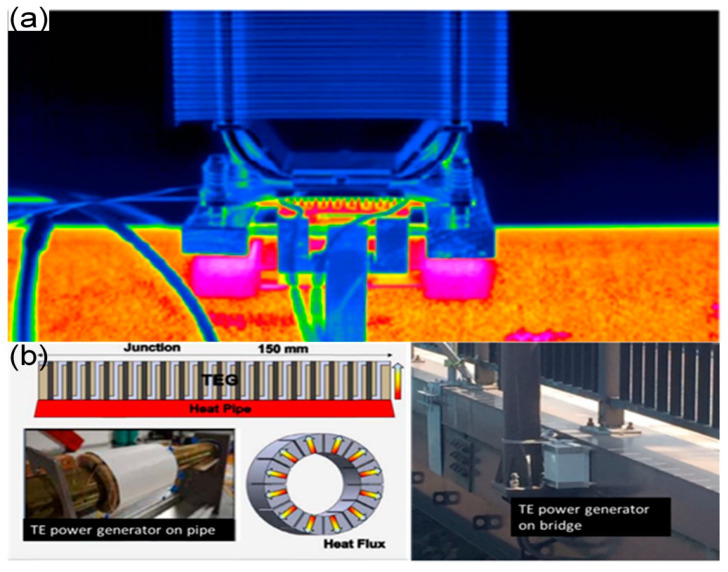
(**a**) Self-powered pipe monitoring—the TEG installed on the warm outer wall of this pipe delivers electrical power for operating sensors, power management systems, and radio sensor technology. Source: olaf.schaefer-welsen@ipm.fraunhofer.de. (**b**) Schematic of a radial TEG integrating thermal pipe insulation. Reprinted with permission from [35]. Copyright 2017, Elsevier.

**Figure 8 micromachines-14-00031-f008:**
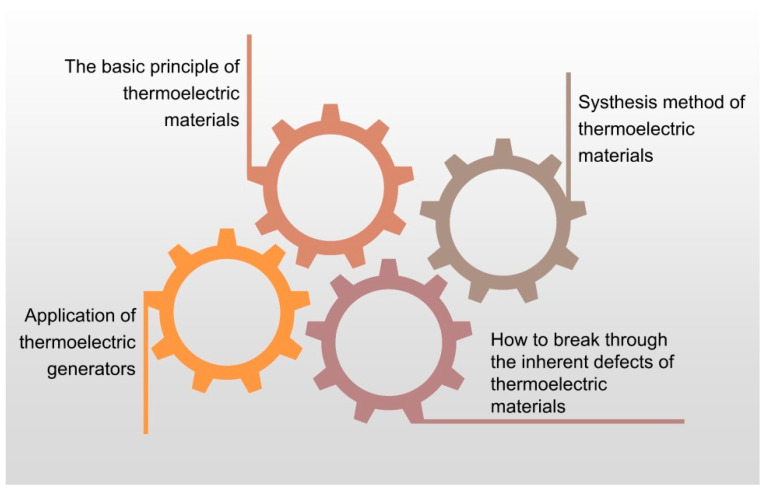
Summary of the article.

**Table 1 micromachines-14-00031-t001:** Power (the output power of TEG (1 mW or sub-1 1 mW) and output voltage of TEG (0.25 V–0.7 V)) required for partial thermoelectric applications.

Application	Power
Wearable watch [34]	100 µW
Human forehead to power a 2-channel EEG [35]	0.8 µW
Energy self-sufficient wireless weather sensor [36]	61.3 µW
Aviation field [37]	30 µW

**Table 2 micromachines-14-00031-t002:** A Summary of TEG applications.

Field	Application
Medical equipment	Wireless dual-channel electroencephalography (EEG) systems. A wireless pulse oximeter. A biomedical hearing aid.
Wireless sensor networks	Combine TEGs with the Bluetooth systems of mobile phones. In aviation applications.
Architecture field	Heating units in buildings are all potential heat sources. The buildings are full of populations that produce large amounts of heat.

## Data Availability

Not applicable.

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
