# Peer review of "Thermoelectric-Powered Sensors for Internet of Things"

_micromachines, 2022, doi:10.3390/mi14010031_

Round 1

Reviewer 1 Report

This article introduces the basic principles, preparation methods, and the supply of electronic devices in different scenarios of thermoelectric materials and thermoelectric devices. This review is systematic and comprehensive, but there are still some deficiencies and mistakes in the introduction of the thermoelectric effect, the preparation and the application scenarios of thermoelectric materials and devices. It is recommended that acceptance be considered after revision. The revision comments about this article are as follows:

1.       In the abstract, the “commonly used” should be revised to “common”, and the “elaborated” is an adjective, and a verb should be added after it to complete the sentence.

2.       In the section of introduction, the formula for conversion efficiency (η) should be added.

   From the perspective of η, higher temperature difference can produce higher conversion efficiency and performance output. There are many methods to increase the temperature difference of the thermoelectric leg, among which optimizing the geometry of the thermoelectric leg can enhance the output performance, you can refer to the following article:

"Novel geometric design of thermoelectric leg based on 3D printing for radioisotope thermoelectric generator." Applied Thermal Engineering 212 (2022): 118514.

3.       In the section of “Synthesis method of thermoelectric materials”, the logic is confusing.

For the thermoelectric materials, the synthesis method refers to the conversion of raw materials (such as metal element or other salts without thermoelectric properties) into thermoelectric materials through chemical or physical reactions. Such as solvothermal method.

For the bulk thermoelectric materials, the “Synthesis method” should be corrected to the “preparation method”. The preparation method refers to preparing the powdered thermoelectric material into a bulk thermoelectric material with a certain shape. There is usually no chemical reaction involved in this process. Such as hot pressing. What’s worse, there is no term of “thermal pressing method”, the correct one is “hot-pressing method”.

For the thermoelectric thin films, the term of “Preparation method” is correct.

In this article, the term of “Preparation method” should apply to both bulk thermoelectric materials and thermoelectric thin films. And the term of “Synthesis method” does not apply to bulk thermoelectric materials. The authors should summarize the preparation methods of bulk thermoelectric materials as well as the synthesis methods of thermoelectric materials, and make revisions to this section.

4.       Please add the solution-based 3D printing to the section of 3.2.4, and refer to the paper in comment 2.

5.       In the section of “Application of thermoelectric generator (TEG) in IoT sensors”, please add the power requirements of the wearable devices, medical equipment and wireless sensor as well as the output voltage and output power generated by the thermoelectric devices in these research works.

Author Response

Reviewer#1 (original comments by reviewers are in blue)

Comment 1: In the abstract, the "commonly used" should be revised to "common", and the "elaborated" is an adjective, and a verb should be added after it to complete the sentence.

In this review, we first outlined the basic principle of the thermoelectricity effect and summarized the common preparation methods for thermoelectric functional parts in TEGs. Then, the application of TEG-powered sensors in the human body was elaborated, including wearable and implantable medical electronic devices.

Comment 2: In the section of introduction, the formula for conversion efficiency (η) should be added.

From the perspective of η, higher temperature difference can produce higher conversion efficiency and performance output. There are many methods to increase the temperature difference of the thermoelectric leg, among which optimizing the geometry of the thermoelectric leg can enhance the output performance, you can refer to the following article:

"Novel geometric design of thermoelectric leg based on 3D printing for radioisotope thermoelectric generator." Applied Thermal Engineering 212 (2022): 118514.

Response: Thanks for your advice. The formula for conversion efficiency (η) has been added in lines 96-103, on pages 3.

“The energy conversion efficiency[1] is determined by the ZT value, and the temperature difference between hot and cold ends as follows:

Where Th represents the temperature of the hot side of TEG, and Tc represents the temperature of the cold side of TEG. The higher temperature difference can produce higher conversion efficiency and performance output. There are many methods to increase the temperature difference of the thermoelectric leg, among which optimizing the geometry of the thermoelectric leg can enhance the output performance.”

Comment 3: In the section of "Synthesis method of thermoelectric materials", the logic is confusing.

For the thermoelectric materials, the synthesis method refers to the conversion of raw materials (such as metal element or other salts without thermoelectric properties) into thermoelectric materials through chemical or physical reactions. Such as solvothermal method.

For the bulk thermoelectric materials, the "Synthesis method" should be corrected to the "preparation method". The preparation method refers to preparing the powdered thermoelectric material into a bulk thermoelectric material with a certain shape. There is usually no chemical reaction involved in this process. Such as hot pressing. What's worse, there is no term of "thermal pressing method", the correct one is "hot-pressing method".

For the thermoelectric thin films, the term of "Preparation method" is correct.

In this article, the term of "Preparation method" should apply to both bulk thermoelectric materials and thermoelectric thin films. And the term of "Synthesis method" does not apply to bulk thermoelectric materials. The authors should summarize the preparation methods of bulk thermoelectric materials as well as the synthesis methods of thermoelectric materials, and make revisions to this section.

Response: Thanks for pointing out the inaccuracy. We realized the difference between the "Synthesis method " and "Preparation method" and corrected them based on your suggestions. Since the " preparation method " applies to both the bulk thermoelectric materials and the thermoelectric thin films, we changed the synthesis method to the preparation method. For the bulk thermoelectric materials, the "Synthesis method" has also been corrected to the "preparation method," and the "thermal pressing method" has been corrected to the "hot-pressing method".

Comment 4: Please add the solution-based 3D printing to the section of 3.2.4, and refer to the paper in comment 2.

Response: Thanks for your advice. We have added the 3D printing technology to 3.2.4. in lines 183-189, on page 5.

"3D printing is a rapid manufacturing technology which is also simple and low cost. It is possible to construct objects layer by layer using molten materials based on digital models. For example, Bian et al.[1] used 3D printing technology to make helical and supporting thermoelectric legs with special shapes and tested their output performance. They found that the customized legs have higher open-circuit voltage and output power than the conventional legs.The schematic figure is added in figure 3d.

Figure 3d. 3D printing technology

Comment 5: In the section of "Application of thermoelectric generator (TEG) in IoT sensors", please add the power requirements of the wearable devices, medical equipment and wireless sensor as well as the output voltage and output power generated by the thermoelectric devices in these research works.

Response: Thanks for your advice. We have added some power requirements from the wearable devices, medical equipment, and wireless sensor, as well as the output voltage and output power generated by the thermoelectric devices in these research works on page 12.

Table 1. Power [1]required for partial thermoelectric applications

Application

Power

Wearable watch[43]

100

Human forehead to power a 2-channel  EEG[42]

0.8

Energy self-sufficient wireless weather sensor[53]

61.3

Aviation field[54]

30

References

1.Bian, M.; Xu, Z.; Meng, C.; Zhao, H.; Tang, X. Novel Geometric Design of Thermoelectric Leg Based on 3D  Printing for Radioisotope Thermoelectric Generator. Appl. Therm. Eng. 2022, 212, 118514,  doi:10.1016/j.applthermaleng.2022.118514.

2.Leonov, V.; Torfs, T.; Fiorini, P.; Van Hoof, C. Thermoelectric Converters of Human Warmth for Self-Powered Wireless Sensor Nodes. IEEE Sens. J. 2007, 7, 650–657, doi:10.1109/JSEN.2007.894917.

3.Thielen, M.; Sigrist, L.; Magno, M.; Hierold, C.; Benini, L. Human Body Heat for Powering Wearable Devices: From Thermal Energy to Application. Energy Convers. Manag. 2017, 131, 44–54, doi:10.1016/j.enconman.2016.11.005.

4.Rösch, A.G.; Gall, A.; Aslan, S.; Hecht, M.; Franke, L.; Mallick, M.M.; Penth, L.; Bahro, D.; Friderich, D.; Lemmer, U. Fully Printed Origami Thermoelectric Generators for Energy-Harvesting. npj Flex. Electron. 2021, 5, 1, doi:10.1038/s41528-020-00098-1.

5.Dilhac, J.-M.; Monthéard, R.; Bafleur, M.; Boitier, V.; Durand-Estèbe, P.; Tounsi, P. Implementation of Thermoelectric Generators in Airliners for Powering Battery-Free Wireless Sensor Networks. J. Electron. Mater. 2014, 43, 2444–2451, doi:10.1007/s11664-014-3150-1.

6.Wang, W.; Cionca, V.; Wang, N.; Hayes, M.; O’Flynn, B.; O’Mathuna, C. Thermoelectric Energy Harvesting for Building Energy Management Wireless Sensor Networks. Int. J. Distrib. Sens. Networks 2013, 9, 232438, doi:10.1155/2013/232438.

7.Wu, T.; Gao, P. Development of Perovskite-Type Materials for Thermoelectric Application. Materials (Basel). 2018, 11 (6), 999. http://dio.org/10.3390/ma11060999.

8.Ye, T.; Wang, X.; Li, X.; Yan, A.Q.; Ramakrishna, S.; Xu, J. Ultra-High Seebeck Coefficient and Low Thermal Conductivity of a Centimeter-Sized Perovskite Single Crystal Acquired by a Modified Fast Growth Method. J. Mater. Chem. C 2017, 5, 1255–1260, doi:10.1039/c6tc04594d.

9.Van Toan, N.; Thi Kim Tuoi, T.; Van Hieu, N.; Ono, T. Thermoelectric Generator with a High Integration Density for Portable and Wearable Self-Powered Electronic Devices. Energy Convers. Manag. 2021, 245, 114571, doi:10.1016/j.enconman.2021.114571.

10.Zeng, X.; Deng, H.T.; Wen, D.L.; Li, Y.Y.; Xu, L.; Zhang, X.S. Wearable Multi-Functional Sensing Technologyfor Healthcare Smart Detection. Micromachines 2022, 13, 1–22.

11.Cao, J.; Zheng, J.; Liu, H.; Tan, C.K.I.; Wang, X.; Wang, W.; Zhu, Q.; Li, Z.; Zhang, G.; Wu, J.; et al. Flexible Elemental Thermoelectrics with Ultra-High Power Density. Mater. Today Energy 2022, 25, 100964, doi:10.1016/j.mtener.2022.100964.

12.Wang, X.; Suwardi, A.; Lim, S.L.; Wei, F.; Xu, J. Transparent Flexible Thin-Film p–n Junction Thermoelectric Module. npj Flex. Electron. 2020, 4, 1–9, doi:10.1038/s41528-020-00082-9.

13.Zhu, S.; Fan, Z.; Feng, B.; Shi, R.; Jiang, Z.; Peng, Y.; Gao, J.; Miao, L.; Koumoto, K. Review on Wearable Thermoelectric Generators: From Devices to Applications. Energies 2022, 15, 3375, doi:10.3390/en15093375.

14.Deng, L.; Liu, Y.; Zhang, Y.; Wang, S.; Gao, P. Organic Thermoelectric Materials: Niche Harvester of Thermal Energy. Adv. Funct. Mater. 2022, 2210770, 2210770, doi:10.1002/adfm.202210770.

15.Lu Z, Zhang H, Mao C,Chang M.; Silk fabric-based wearable thermoelectric generator for energy harvesting from the human body[J]. Applied energy, 2016, 164: 57-63.

16.Sanislav, T.; Mois, G.D.; Zeadally, S.; Folea, S.C. Energy Harvesting Techniques for Internet of Things (IoT). IEEE Access 2021, 9, 39530–39549, doi:10.1109/ACCESS.2021.3064066.

[1] The output power of TEG(1mW or sub-1 1mW) and output voltage of TEG (0.25V-0.7V)[6]

Reviewer 2 Report

TE generator shows the great potential application in different areas, such as the authors discussed the Internet-of-Things (IoT) combines various sensors and the internet to form an expanded network, realizing the interconnection between human beings and machines anytime and anywhere. However, the authors do not reflect this in the manuscript, but rather in a cursory, broad discussion of thermoelectric materials and device preparation for some applications. There is no focus on IoT applications, advantages and disadvantages, and an overview of future developments. In fact, the title and the article of this manuscript are inconsistent.

1. For TE materials, the perovskite materials crystal and thin film are also great potential materials, but not included here. (DOI: 10.1039/C6TC04594D (Paper) J. Mater. Chem. C, 2017, 5, 1255-1260)

2. TE device can be used as a sensor and power source for IoT, this should be one point in this manuscript, but the authors did not do it. 

3. For thin film TE device, there is some real efficiency device reported and can use for IoT application, such like: (https://doi.org/10.1016/j.mtener.2022.100964,  https://doi.org/10.1038/s41528-020-00082-9 )

4. After discussing the various applications, the authors do not give a complete chapter and paragraph discussing the network for the practical application of TE in IoT. This is a missing point in the focus of the article.

Author Response

Reviewer#2

  Comment 1: For TE materials, the perovskite materials crystal and thin film are also great potential materials, but not included here. (DOI: 10.1039/C6TC04594D (Paper) J. Mater. Chem. C, 2017, 5, 1255-1260)

Response: Thanks for your suggestion. We have added the introduction of perovskite material to the manuscript in lines 278-284, on page 7.

"Perovskite materials have attracted much attention in thermoelectric applications due to their low thermal conductivity, high carrier mobility, and the Seebeck coefficient. Ye et al.[8] developed organic-inorganic lead halide perovskite single crystal (MAPbI3) based TE devices and found that improving electrical conductivity is the key factor to realize the high thermal electric performance of perovskite."

Comment 2: TE device can be used as a sensor and power source for IoT, this should be one point in this manuscript, but the authors did not do it.

Response: Thanks for your advice. TEG for IoT power source is one of the most promising applications. Therefore, we added relative paragraphs on using thermoelectric materials as power sources and sensors for the Internet of Things in lines 55-62, on page 2.

"Thermoelectric devices, with their miniature design and can produce orders of magnitude higher energy than the chemical energy produced by the same amount of material, are the perfect power source and sensors for IoT applications. Nguyen et al.[9] used thermoelectric materials as a power source for the calculator and electronic watch. Zeng et al.[10] considered using thermoelectric sensors in wearable medical devices to detect physiological conditions such as breathing, pulse, etc. We mainly introduce the application of thermoelectric materials in the IoTs from the aspect of thermoelectric materials as sensors."

Comment 3: For thin film TE device, there is some real efficiency device reported and can use for IoT application, such like: (https://doi.org/10.1016/j.mtener.2022.100964, https://doi.org/10.1038/s41528-020-00082-9 )

Response: Thanks for your advice. We have added the references on thin-film devices and related discussion about their potential application in the Internet of Things on page 7.

"Flexible thermoelectric materials show broad prospects in wearable thermoelectric materials. To avoid the low performance of organic thermoelectric films, Cao et al.[11] prepared flexible thermoelectric films by sputtering Sb (p-type) and Ni (n-type) films on the polyimide substrates. Since transparency is essential for wearable devices. Wang et al.[12] prepared TEG with p-n junctions formed by p-type PEDOT: PSS and n-type indium tin oxide (ITO), respectively, to obtain transparent flexible thermoelectric films with high conductivity and Seebeck coefficient."

Comment 4: After discussing the various applications, the authors do not give a complete chapter and paragraph discussing the network for the practical application of TE in IoT. This is a missing point in the focus of the article.

Response: Thanks for your advice. We have summarized the application of TEGs in the Internet of Things on page 12.

"In conclusion, the wearable devices powered by mini-TEGs for monitoring the medical or physical conditions of patients need to form wireless sensor networks to realize real-time detection and diagnosis of the status of the objects. Furthermore, by taking TEGs as the energy supply for signal transmission and reception units, the advantages will be the ability to reuse the widely available low-grade thermoenergy from the human body or buildings, which can alleviate the extra cost from the external energy supply. A summary of TEG applications for IoT is listed in Table 2."

Table 2. A Summary of TEG applications

Field

Application

Medical equipment

1. wireless dual-channel electroencephalography (EEG) systems

2. a wireless pulse oximeter

3. a biomedical hearing aid        

Wireless sensor networks

1. combine TEGs with the Bluetooth system of mobile phones

2. in aviation applications

Architecture field

1. heating units in buildings are all potential heat sources

2. The buildings are full of populations produced a large amount of heat  

References

1.Bian, M.; Xu, Z.; Meng, C.; Zhao, H.; Tang, X. Novel Geometric Design of Thermoelectric Leg Based on 3D  Printing for Radioisotope Thermoelectric Generator. Appl. Therm. Eng. 2022, 212, 118514,  doi:10.1016/j.applthermaleng.2022.118514.

2.Leonov, V.; Torfs, T.; Fiorini, P.; Van Hoof, C. Thermoelectric Converters of Human Warmth for Self-Powered Wireless Sensor Nodes. IEEE Sens. J. 2007, 7, 650–657, doi:10.1109/JSEN.2007.894917.

3.Thielen, M.; Sigrist, L.; Magno, M.; Hierold, C.; Benini, L. Human Body Heat for Powering Wearable Devices: From Thermal Energy to Application. Energy Convers. Manag. 2017, 131, 44–54, doi:10.1016/j.enconman.2016.11.005.

4.Rösch, A.G.; Gall, A.; Aslan, S.; Hecht, M.; Franke, L.; Mallick, M.M.; Penth, L.; Bahro, D.; Friderich, D.; Lemmer, U. Fully Printed Origami Thermoelectric Generators for Energy-Harvesting. npj Flex. Electron. 2021, 5, 1, doi:10.1038/s41528-020-00098-1.

5.Dilhac, J.-M.; Monthéard, R.; Bafleur, M.; Boitier, V.; Durand-Estèbe, P.; Tounsi, P. Implementation of Thermoelectric Generators in Airliners for Powering Battery-Free Wireless Sensor Networks. J. Electron. Mater. 2014, 43, 2444–2451, doi:10.1007/s11664-014-3150-1.

6.Wang, W.; Cionca, V.; Wang, N.; Hayes, M.; O’Flynn, B.; O’Mathuna, C. Thermoelectric Energy Harvesting for Building Energy Management Wireless Sensor Networks. Int. J. Distrib. Sens. Networks 2013, 9, 232438, doi:10.1155/2013/232438.

7.Wu, T.; Gao, P. Development of Perovskite-Type Materials for Thermoelectric Application. Materials (Basel). 2018, 11 (6), 999. http://dio.org/10.3390/ma11060999.

8.Ye, T.; Wang, X.; Li, X.; Yan, A.Q.; Ramakrishna, S.; Xu, J. Ultra-High Seebeck Coefficient and Low Thermal Conductivity of a Centimeter-Sized Perovskite Single Crystal Acquired by a Modified Fast Growth Method. J. Mater. Chem. C 2017, 5, 1255–1260, doi:10.1039/c6tc04594d.

9.Van Toan, N.; Thi Kim Tuoi, T.; Van Hieu, N.; Ono, T. Thermoelectric Generator with a High Integration Density for Portable and Wearable Self-Powered Electronic Devices. Energy Convers. Manag. 2021, 245, 114571, doi:10.1016/j.enconman.2021.114571.

10.Zeng, X.; Deng, H.T.; Wen, D.L.; Li, Y.Y.; Xu, L.; Zhang, X.S. Wearable Multi-Functional Sensing Technologyfor Healthcare Smart Detection. Micromachines 2022, 13, 1–22.

11.Cao, J.; Zheng, J.; Liu, H.; Tan, C.K.I.; Wang, X.; Wang, W.; Zhu, Q.; Li, Z.; Zhang, G.; Wu, J.; et al. Flexible Elemental Thermoelectrics with Ultra-High Power Density. Mater. Today Energy 2022, 25, 100964, doi:10.1016/j.mtener.2022.100964.

12.Wang, X.; Suwardi, A.; Lim, S.L.; Wei, F.; Xu, J. Transparent Flexible Thin-Film p–n Junction Thermoelectric Module. npj Flex. Electron. 2020, 4, 1–9, doi:10.1038/s41528-020-00082-9.

13.Zhu, S.; Fan, Z.; Feng, B.; Shi, R.; Jiang, Z.; Peng, Y.; Gao, J.; Miao, L.; Koumoto, K. Review on Wearable Thermoelectric Generators: From Devices to Applications. Energies 2022, 15, 3375, doi:10.3390/en15093375.

14.Deng, L.; Liu, Y.; Zhang, Y.; Wang, S.; Gao, P. Organic Thermoelectric Materials: Niche Harvester of Thermal Energy. Adv. Funct. Mater. 2022, 2210770, 2210770, doi:10.1002/adfm.202210770.

15.Lu Z, Zhang H, Mao C,Chang M.; Silk fabric-based wearable thermoelectric generator for energy harvesting from the human body[J]. Applied energy, 2016, 164: 57-63.

16.Sanislav, T.; Mois, G.D.; Zeadally, S.; Folea, S.C. Energy Harvesting Techniques for Internet of Things (IoT). IEEE Access 2021, 9, 39530–39549, doi:10.1109/ACCESS.2021.3064066.

Reviewer 3 Report

The manuscript "Thermoelectric-Powered Sensors for Internet-of-Things" by Xie et al. presents a review on the mechanism of thermoelectric generator (TEG), commonly used thermoelectric materials and its preparation methods, as well as TEG devices towards the application of wearables, implantable medical electronic devices for Internet-of-things. Overall, this research is well-organized and will be of interest to a broad readership. However, a few important clarifications need to be addressed before the manuscript can be considered for publication. The comments that need to be addressed are as follows:

1) The authors stated in the abstract, lines 7-8, that “In this review, we first summarize the basic principle of thermoelectricity and overview the types of commonly used thermoelectric materials and their preparation methods.” However, the working mechanism of the TEG and novel thermoelectric materials have not been discussed. The TEG mechanism (including voltage and power generation equations) and thermoelectric materials (applicable to the applications discussed) should be briefly discussed.

2) The authors present TEG applications for wearable, medical devices, and sensors. However, they do not discuss biocompatibility, toxicity, washability, and wearability, as well as any of the other practical challenges of TEGs that must be addressed to make it feasible for such applications.

3) The authors have highlighted several important applications in the review. However, the power demands of the devices (wearable and medical devices, wireless sensor networks) and the power outputs provided by the TEG have not been clearly mentioned. It is remarkably important to discuss it.

4) Most of the TEG applications have already been discussed. Add novel applications to the appropriate application sections.

Author Response

Reviewer#3

Comment 1: The authors stated in the abstract, lines 7-8, that "In this review, we first summarize the basic principle of thermoelectricity and overview the types of commonly used thermoelectric materials and their preparation methods." However, the working mechanism of the TEG and novel thermoelectric materials have not been discussed. The TEG mechanism (including voltage and power generation equations) and thermoelectric materials (applicable to the applications discussed) should be briefly discussed in lines 87-90、101-104, page 3.

Response: Thanks for your advice. The TEG mechanism and thermoelectric materials have been discussed in lines 82-85, 96-103, and on page 3.

 "Materials have different concentrations of free carriers at different temperatures. Therefore, when there is a temperature difference between both ends of the thermoelectric material, the thermodynamically-driven directional diffusion of the electron can form a potential difference, called the Seebeck voltage, as given in Equation: V = S × ∆T[7]".

“The energy conversion efficiency[1,14] is determined by the ZT value, and the temperature difference between hot and cold ends as follows:

Where Th represents the temperature of the hot side of TEG, and Tc represents the temperature of the cold side of TEG. The higher temperature difference can produce higher conversion efficiency and performance output. There are many methods to increase the temperature difference of the thermoelectric leg, among which optimizing the geometry of the thermoelectric leg can enhance the output performance.”

Comment 2: The authors present TEG applications for wearable, medical devices, and sensors. However, they do not discuss biocompatibility, toxicity, washability, and wearability, as well as any of the other practical challenges of TEGs that must be addressed to make it feasible for such applications.

Response: Thanks for pointing out the insufficiency. We have briefly discussed the shortcomings of thermoelectric in the actual application scenarios on page 7.

"When using TEG as the power supply for wearable equipment, in addition to the requirement of conversion efficiency, one has to consider the additional requirements, such as biocompatibility, wear resistance, toxicity, flexibility, and so on. So far, Bi2Te3 and its derivatives are the most commonly used low-temperature thermoelectric materials, showing high thermoelectric performance around room temperature. Although the advantage of mass production, they suffered from poor mechanical properties and toxicity of telluride, which limits its application in some wearable or implanted devices. In general, the current popular thermoelectric materials are not yet satisfactory for powering wearable sensor devices, considering their cost and limited large-scale availability with high quality. For in-vitro application, Lu et al.[15] deposited the nanostructured telluride on silk fabric to produce a flexible thermoelectric material, which could effectively convert the body heat energy into electricity. The reduced direct contact with the human body by changing the position of telluride on fabrics can reduce the harm to the human body."

Comment 3: The authors have highlighted several important applications in the review. However, the power demands of the devices (wearable and medical devices, wireless sensor networks) and the power outputs provided by the TEG have not been clearly mentioned. It is remarkably important to discuss it.

Response: Thanks for pointing out the insufficiency. We have discussed and listed the specific TEG output power and the power required for the equipment in table 1 on page 7.

     Table 1. Power[1] required for partial thermoelectric applications

Application

Power

Wearable watch[43]

100

Human forehead to power a 2-channel  EEG[42]

0.8

Energy self-sufficient wireless weather sensor[53]

61.3

Aviation field[54]

30

Comment 4: Most of the TEG applications have already been discussed. Add novel applications to the appropriate application sections.

Response: Thanks for your advice. Thermoelectric applications in other areas have been added in lines 434-436, on page 14.

"Of course, beyond the application areas of thermoelectric materials mentioned above, they have also been used in environmental monitoring, civil infrastructure, national defense, manufacturing, and other applications.[16]

References

1.Bian, M.; Xu, Z.; Meng, C.; Zhao, H.; Tang, X. Novel Geometric Design of Thermoelectric Leg Based on 3D  Printing for Radioisotope Thermoelectric Generator. Appl. Therm. Eng. 2022, 212, 118514,  doi:10.1016/j.applthermaleng.2022.118514.

2.Leonov, V.; Torfs, T.; Fiorini, P.; Van Hoof, C. Thermoelectric Converters of Human Warmth for Self-Powered Wireless Sensor Nodes. IEEE Sens. J. 2007, 7, 650–657, doi:10.1109/JSEN.2007.894917.

3.Thielen, M.; Sigrist, L.; Magno, M.; Hierold, C.; Benini, L. Human Body Heat for Powering Wearable Devices: From Thermal Energy to Application. Energy Convers. Manag. 2017, 131, 44–54, doi:10.1016/j.enconman.2016.11.005.

4.Rösch, A.G.; Gall, A.; Aslan, S.; Hecht, M.; Franke, L.; Mallick, M.M.; Penth, L.; Bahro, D.; Friderich, D.; Lemmer, U. Fully Printed Origami Thermoelectric Generators for Energy-Harvesting. npj Flex. Electron. 2021, 5, 1, doi:10.1038/s41528-020-00098-1.

5.Dilhac, J.-M.; Monthéard, R.; Bafleur, M.; Boitier, V.; Durand-Estèbe, P.; Tounsi, P. Implementation of Thermoelectric Generators in Airliners for Powering Battery-Free Wireless Sensor Networks. J. Electron. Mater. 2014, 43, 2444–2451, doi:10.1007/s11664-014-3150-1.

6.Wang, W.; Cionca, V.; Wang, N.; Hayes, M.; O’Flynn, B.; O’Mathuna, C. Thermoelectric Energy Harvesting for Building Energy Management Wireless Sensor Networks. Int. J. Distrib. Sens. Networks 2013, 9, 232438, doi:10.1155/2013/232438.

7.Wu, T.; Gao, P. Development of Perovskite-Type Materials for Thermoelectric Application. Materials (Basel). 2018, 11 (6), 999. http://dio.org/10.3390/ma11060999.

8.Ye, T.; Wang, X.; Li, X.; Yan, A.Q.; Ramakrishna, S.; Xu, J. Ultra-High Seebeck Coefficient and Low Thermal Conductivity of a Centimeter-Sized Perovskite Single Crystal Acquired by a Modified Fast Growth Method. J. Mater. Chem. C 2017, 5, 1255–1260, doi:10.1039/c6tc04594d.

9.Van Toan, N.; Thi Kim Tuoi, T.; Van Hieu, N.; Ono, T. Thermoelectric Generator with a High Integration Density for Portable and Wearable Self-Powered Electronic Devices. Energy Convers. Manag. 2021, 245, 114571, doi:10.1016/j.enconman.2021.114571.

10.Zeng, X.; Deng, H.T.; Wen, D.L.; Li, Y.Y.; Xu, L.; Zhang, X.S. Wearable Multi-Functional Sensing Technologyfor Healthcare Smart Detection. Micromachines 2022, 13, 1–22.

11.Cao, J.; Zheng, J.; Liu, H.; Tan, C.K.I.; Wang, X.; Wang, W.; Zhu, Q.; Li, Z.; Zhang, G.; Wu, J.; et al. Flexible Elemental Thermoelectrics with Ultra-High Power Density. Mater. Today Energy 2022, 25, 100964, doi:10.1016/j.mtener.2022.100964.

12.Wang, X.; Suwardi, A.; Lim, S.L.; Wei, F.; Xu, J. Transparent Flexible Thin-Film p–n Junction Thermoelectric Module. npj Flex. Electron. 2020, 4, 1–9, doi:10.1038/s41528-020-00082-9.

13.Zhu, S.; Fan, Z.; Feng, B.; Shi, R.; Jiang, Z.; Peng, Y.; Gao, J.; Miao, L.; Koumoto, K. Review on Wearable Thermoelectric Generators: From Devices to Applications. Energies 2022, 15, 3375, doi:10.3390/en15093375.

14.Deng, L.; Liu, Y.; Zhang, Y.; Wang, S.; Gao, P. Organic Thermoelectric Materials: Niche Harvester of Thermal Energy. Adv. Funct. Mater. 2022, 2210770, 2210770, doi:10.1002/adfm.202210770.

15.Lu Z, Zhang H, Mao C,Chang M.; Silk fabric-based wearable thermoelectric generator for energy harvesting from the human body[J]. Applied energy, 2016, 164: 57-63.

16.Sanislav, T.; Mois, G.D.; Zeadally, S.; Folea, S.C. Energy Harvesting Techniques for Internet of Things (IoT). IEEE Access 2021, 9, 39530–39549, doi:10.1109/ACCESS.2021.3064066.

[1] The output power of TEG(1mW or sub-1 1mW) and output voltage of TEG (0.25V-0.7V)

Round 2

Reviewer 1 Report

All questions have been answered and the paper can be published.

Reviewer 2 Report

Agree to accept